# Paracrine Responses of Cardiosphere-Derived Cells to Cytokines and TLR Ligands: A Comparative Analysis

**DOI:** 10.3390/ijms242417278

**Published:** 2023-12-08

**Authors:** Ekaterina Zubkova, Konstantin Dergilev, Irina Beloglazova, Alexander Kalinin, Alika Guseva, Alexander Andreev, Stanislav Partigulov, Mikhail Lepilin, Mikhail Menshikov, Yelena Parfyonova

**Affiliations:** 1National Medical Research Centre of Cardiology Named after Academician E.I. Chazov, 121552 Moscow, Russia; cat.zubkova@gmail.com (E.Z.); alexanderpkalinin@gmail.com (A.K.); guseva@cardio.ru (A.G.); myumensh@mail.ru (M.M.); 2The Faculty of Fundamental Medicine, Lomonosov Moscow State University, 119991 Moscow, Russia

**Keywords:** cardiosphere-derived cells, cytokines, TLRs, secretome

## Abstract

Cardiosphere-derived cells (CDCs) are currently being evaluated in clinical trials as a potential therapeutic tool for regenerative medicine. The effectiveness of transplanted CDCs is largely attributed to their ability to release beneficial soluble factors to enhance therapeutic effects. An emerging area of research is the pretreatment of stem cells, including CDCs, with various cytokines to improve their therapeutic properties. This strategy aims to enhance their survival, proliferation, differentiation, and paracrine activities after transplantation. In our study, we investigated the differential effects of various cytokines and TLR ligands on the secretory phenotype of human CDCs. Using a magnetic bead-based immunoassay, we analyzed the CDCs-conditioned media for 41 cytokines and growth factors and detected the presence of 21 cytokines. We found that CDC incubation with lipopolysaccharide, a TLR4 ligand, and the cytokine combination of TNF/IFN significantly increased the secretion of most of the cytokines detected. Specifically, we observed an increased secretion and gene expression of IP10, MCP3, IL8, and VEGFA. In contrast, the TLR3 ligand polyinosinic-polycytidylic acid and TGF-beta had minimal effects on CDC cytokine secretion. Additionally, TNF/IFN, but not LPS, enhanced ICAM1 expression. Our findings offer new insights into the role of cytokines in potentially modulating the biology and regenerative potential of CDCs.

## 1. Introduction

Cell therapy using stem/progenitor cells has opened new avenues in regenerative medicine, offering opportunities for the treatment of a wide range of diseases. Specifically, mesenchymal stromal cells (MSCs) possess both regenerative and immunomodulatory potential, making them viable candidates for treating various pathologies, ranging from immunological disorders to diseases of the central nervous system and heart. Particularly noteworthy are cardiac cardiosphere-derived cells (CDCs), which differ in their properties from MSCs originating from other sources and may offer higher reparative potential for heart-related conditions [1,2,3,4].

CDCs are phase-bright, small, round cardiac explant-derived cells that spontaneously aggregate under low-adhesion conditions [5,6]. They comprise a heterogeneous population of stromal vascular cardiac cells, including resident MSCs, pericytes, fibro-blasts, and progenitor cells [6,7]. Due to their ability to mimic native cardiac niches [8], CDCs serve as a unique model system for investigating reparative and re-generative processes following cardiac damage. 

The growing body of literature emphasizes the unique cellular composition of CDCs and their ability to secrete a distinctive set of biologically active compounds that are both soluble and in extracellular vesicles/exosomes. Human clinical trials such as ALLSTAR, DY-NAMIC, HOPE-Duchenne, and HOPE-2 have confirmed the disease-modifying bioactivity of CDCs in humans. The now widely accepted “paracrine hypothesis” suggests that the effectiveness of transplanted CDC cells is primarily based on their ability to secrete beneficial soluble factors, such as cytokines, growth factors, and microRNAs, to initiate and promote therapeutic effects [9,10].

A significant role in the regulation of CDC secretome is played by inflammation-related signals [11,12,13]. During the acute phase of myocardial infarction, a pro-inflammatory environment is formed under the influence of free radicals and cytokines [14]. This environment serves dual purposes: activating and accumulating cells for debris resorption and preparing for the proliferative healing phase. As such, post-infarction heart repair involves a transition from a pro- to an anti-inflammatory environment, modulated by various mediators (cytokines, growth factors, exosomes/microvesicles, microRNAs) affecting all cell types involved in heart damage and repair [15].

Cytokines secreted at different stages during cardiac inflammation can potentially influence the phenotype of CDCs and shape their secretome, offering exciting therapeutic implications. While the application of stromal cells for tissue regeneration is a promising approach, it encounters several significant challenges. Factors such as oxidative stress, acidosis, hypoxia, inflammation, and the inadequate supply of nutrients and oxygen in the injured area create a hostile microenvironment, hindering the efficient participation of stromal cells in the regeneration process. To address these challenges, a strategy known as “cell preconditioning” was proposed. This approach serves a dual purpose: firstly, it amplifies specific beneficial cell signaling pathways; and secondly, it exposes cells to stress conditions resembling the harsh environment they will encounter in the injured area. The goal of this method is to prepare cells to adapt effectively to the hostile host environment before administration [16]. 

Extensive research into the preconditioning of MSCs, including exposure to hypoxia or the use of cytokines, pharmacological and chemical agents have been applied to enhance the overall efficacy of MSC transplantation. All of these techniques promote MSC survival and their capacity to form a regenerative and proliferative environment [17]. However, comprehensive investigations into the preconditioning of CDCs remain relatively limited, although certain studies have indicated the beneficial effects of preconditioning CDCs with hypoxia and H_2_O_2_ [18,19].

Specifically, the roles of TGF-beta and IL4, cytokines traditionally associated with anti-inflammatory responses, and the INF-gamma/TNF-alpha combination, which has shown promise in enhancing the immunosuppressive capabilities of MSCs, are not well understood in the context of CDCs. Therefore, we hypothesize that treatment of CDCs with TGF-beta and IL4, as well as a combination of INF-gamma and TNF-alpha, may induce phenotypes changes in CDCs. 

Furthermore, we also included Toll-like receptors (TLR) ligands—lipopolysaccharide (LPS) and polyinosinic:polycytidylic acid (poly(I:C)) in our work—based on results from the Betancourt group. Their research has suggested that these ligands may polarize mesenchymal stromal cells into pro-inflammatory and anti-inflammatory phenotypes, akin to the M1 and M2 macrophage polarization [20,21]. 

Our study aims to deepen our understanding of how cytokine pretreatment can modulate the secretome of CDCs, which may, in turn, enhance their therapeutic potential. Our findings lay the groundwork for future in vivo studies that will directly assess the efficacy of such pretreatment strategies in the context of cardiac regenerative medicine.

## 2. Results

### 2.1. The Phenotype and Culture Characteristics of CDCs

Adult cardiac stem/progenitor cells were isolated and expanded using the traditional method. After mincing and partial enzymatic digestion, the patients’ biopsy fragments were seeded on fibronectin-coated plates as explants. Cardiosphere-forming cells, which migrated from the explants, were initially harvested 8 or more days after obtaining the specimen. After seeding cells on poly(2-hydroxyethylmethacrylate) (poly-HEMA)-precoated plates, cardiospheres spontaneously formed within 48 h. Representative cardiospheres are illustrated in Figure 1a (top).

Subsequently, CDCs underwent two passages as adherent monolayers, after which they were used for flow cytometry and cell proliferation experiments. The characteristic morphology of CDCs during expansion at passage 2 can be observed in Figure 1a (bottom). Through a flow cytometry analysis utilizing forward scatter (FSC) for size and side scatter (SSC) for granularity, we observed the following patterns on the graph: MSCs are positioned more towards the top right, indicating larger size and higher granularity. HUVECs, in contrast, demonstrate smaller size and reduced granularity compared with MSCs. CDCs display characteristics that are intermediate between MSCs and HUVECs (Figure 1b).

Despite their widespread clinical use, the phenotype of CDCs remains underexplored, and their cellular origin in the heart is still ambiguous. Numerous studies have shown that CDCs are uniformly CD105-positive (CD105—a regulatory component of the transforming growth factor-β receptor complex significant in angiogenesis [22]) and CD45-negative (a pan-hematopoietic marker). Unlike fibroblasts, CDCs have low CD90 expression. However, the expression of CD90 has been variable, ranging from 25% [23] to over 60% [24]. There are also inconsistencies regarding the expression of endothelial marker CD31 or c-KIT [25]. In our experiments, all CDCs were positive for CD29, CD73, and CD105. However, only 40–50% of CDCs stained for CD90 (Figure 1c), and no more than 1% stained positive for CD31 (data is contained within Appendix A). These results align well with the findings of Kogan PS et al. [26], in which CD90 expression was 49% and CD31 expression was 0.86%.

Furthermore, we assessed the influence on CDCs viability cytokines (IL4, TGF-beta, and TNF/Interferon) as well as on a TLR4 ligand (LPS) and TLR3 ligand (poly(I:C)). These ligands were included in our study based on the Betancourt group works, which suggested that TLR4 and TLR3 ligands could polarize mesenchymal stem cells into pro-inflammatory and anti-inflammatory phenotypes, respectively, analogous to M1 and M2 macrophages [20,21]. 

We evaluated cell growth over 72 h in the presence of cytokines and TLR ligands (Figure 1d) and (in separate experiments) 72 h after a preliminary 48 h incubation with cytokines (Appendix A). The presented results showed that none of the tested pro- or anti-inflammatory cytokines and TLR ligands had a significant effect on CDCs proliferation.

### 2.2. The Effects of Cytokines and TLR Ligands on Cytokine Expression Pattern of CDCs

The strategy of preconditioning cardiosphere-derived cells (CDCs) with various cytokines to augment their therapeutic efficacy is an emerging research frontier. Given that the paracrine function of CDCs is pivotal in tissue repair [27,28], the molecular basis and components of the CDC secretome are of intense research interest. In this study, we evaluated how pre-incubating CDCs with different cytokines and TLR ligands might modulate their secretion profile.

We specifically selected cytokines that are expressed during different phases of cardiac inflammation. TNF-alpha and IFN-gamma are among the earliest cytokines to be upregulated. TGF-beta typically becomes prominent several days post-injury, persisting throughout the healing phase, and IL4 is more prevalent during the chronic phase, aiding in the resolution of inflammation and tissue remodeling. In accordance with the concept proposed by the Betancourt group, the TLR3 ligand poly(I:C) has the potential to induce an anti-inflammatory phenotype, while TLR4 ligand LPS may induce a pro-inflammatory phenotype in MSCs. However, it is important to note that these ligands’ actions in the context of CDCs have not been extensively explored.

We analyzed 24 h conditioned media via a magnetic bead-based immunoassay using an HCYTMAG-60K-PX41 kit (Merck). Out of the 41 soluble factors that the assay is capable of detecting, we identified 21 that had concentrations above the detection limit. Specifically, the concentrations of TGF-alpha, FLT-3L, INF-alpha2, INF-gamma, IL10, IL-12P40, IL-12P70, IL17A, IL1-alpha, IL1-beta, IL2, IL3, IL5, TNF-alpha, and TNF-beta were below detectable levels. The levels of cytokines such as MDC, sCD40L, IL13, IL9, and IL7 were very low at approximately the first standard of the calibration curve (2–4 pg/mL). Given the low concentrations observed, the reliability of measurements for these specific cytokines may be questionable.

A multiplex analysis of the CDCs’ secretome following incubation with both inflammatory and anti-inflammatory factors revealed that the most pronounced changes were elicited by LPS and the TNF/INF combination. It is important to note that no significant differences in response were detected across the administered doses, whether for the TGF, INF, or LPS treatments.

The summarized secretion profiles from the CDCs of three different donors under the influence of these cytokines are depicted as a heatmap in Figure 2a. 

We found that LPS exposure led to increased secretion of numerous factors, including G-CSF, GM-CSF, GRO1, MCP1, MCP3, IL15, IL6, IL8, IP10, MIP1a, MIP1b, RANTES, and VEGFA (Figure 2a). These effects are likely mediated through the activation of the TLR4 receptor and downstream activation of NF-kB and AP1 factors [29] (Figure 2b,c).

In contrast, another TLR ligand, poly(I:C) acid, which activates TLR3, did not significantly influence the secretion of the analyzed factors (Figure 2b,c). 

Our initial assumption was that MSCs and CDCs would exhibit similar responses in secretion changes following TLR ligand stimulation. However, it turned out that CDCs significantly increase the secretion of a range of factors in response to LPS, namely G-CSF, GM-CSF, MCP3, IP10, MIP1a, RANTES/CCL5, and PDGFAA compared with MSCs. In the case of the TLR3 ligand poly(I:C), no significant difference in impact was noted (Appendix A).

Transforming growth factor (TGFb) only augmented the secretion of growth factors PDGFAA and VEGFA. Anti-inflammatory cytokine IL4, whose effects are mediated by STAT6, enhanced the secretion of eotaxin1 (CCL11), PDGFAA, and IL15. Interferon-gamma boosted the secretion of MCP3, IL15, IP10, and CCL5 while suppressing IL6 and MCP1 secretion (Figure 2b,c). The cytokine combination TNF/INF notably altered the secretion of G-CSF, GM-CSF, GRO1, MCP3, IL15, IL8, IP10, MIP1a, MIP1b, CCL5/RANTES, and VEGF A (Figure 2a).

Our findings underscore that pro-inflammatory cytokines primarily elevate the secretion of monocyte differentiation and attraction (MCP1, MCP3, G-CSF, GM-CSF, CCL5, IP10, MIP1a) and neutrophil recruitment factors (IL8, GRO1), highlighting the role of CDCs in modulating the inflammatory milieu during regeneration. In contrast, cytokines with mainly anti-inflammatory actions, such as IL4 and TGF-beta, deviate from the effects of TNFa/INFg primarily by enhancing PDGFAA secretion, which plays an important role in cardiac repair and in maintaining cardiac connective tissue [30]. We hypothesize that modulating the CDC secretome under anti-inflammatory cytokines might favor the regenerative phase of cardiac repair. It is worth noting that both pro-inflammatory and anti-inflammatory cytokines increased VEGFA secretion to a greater or lesser extent.

### 2.3. Effects of Cytokines and TLR Ligands on CDC Gene Expression

The results from the secretome analysis closely align with the mRNA expression data obtained via a real-time PCR analysis (Figure 3a,b). This evidence underscores the dynamic alterations in the CDCs’ expression profiles, leading to the release of not only stored growth factors and cytokines, but also their de novo synthesis. Furthermore, we identified an upregulation of IL1 beta mRNA, even though we could not assess its concentration in the conditioned medium as it was below the detectable level.

Contrary to the findings obtained from the magnetic bead assay, the mRNA expressions of IL6 and G-CSF showed significant inter-donor variability and, as such, did not achieve statistical significance. We also detected a clear trend toward increased expression of the pro-inflammatory transcription factor STAT1 and a significant increase in the expression of PAI2 and ICAM1 upon exposure to LPS and the combination of TNF/INF. The enhanced expression of ICAM1 is further substantiated by the flow cytometry data (Figure 3c). Meanwhile, intracellular staining with antibodies to PAI2 did not reveal a significant increase in its content. The surface expression of integrin beta1 (CD29), due to high inter-donor variability, showed only a trend toward enhancement. This aligns with the findings from Ezoe, K. and Horikoshi, T [31] on dermal fibroblasts, where TNF-alpha augmented the expression of integrin beta1 by approximately 10%, albeit not significantly.

Some genes under the influence of the TNF/INF combination altered their expression so dramatically that representing the data graphically became challenging. Hence, these data are summarized in Table 1.

Incubation of CDCs with LPS and TNF/INF notably upregulated the expression of type II MHC (*HLA-DRB1*) (Table 1), hinting that cardiosphere cells, like MSCs, may possess antigen-presenting capabilities. Simultaneously, there is a pronounced increase in the expression of immunosuppressive genes, such as PD-L1 (CD274) and IDO1/2. These are widely recognized as major participants of MSCs’ immunomodulatory abilities [32]. We evaluated the impact of CDCs on lymphocyte proliferation induced by phytohemagglutinin (PHA), both in a non-contact Transwell^TM^ co-culture system and in a direct contact co-culture. In the absence of direct intercellular interactions, we observed no changes in lymphocyte proliferation rates (data is contained within Appendix A), emphasizing the pivotal role of contact cell–cell interactions. The direct co-culture of CDCs with PHA-activated lymphocytes resulted in a 10–12% reduction in cells reaching their maximum number of divisions by the time of analysis. CDCs pre-incubated with IL4, TGF-beta, p(I:C), or LPS exerted a similar inhibitory effect on lymphocyte proliferation. However, CDCs pre-incubated with TNF/INF reduced the number of lymphocytes that underwent the maximum number of divisions by approximately 20–25%, an 8–10% increase compared with the control (Appendix A). 

Given that the TNF/INF combination greatly boosts the expression of PD-L1 (CD274) and ICAM1 and considering that ICAM1 has been previously identified as pivotal for MSC-mediated immunosuppression [33] and likely disrupts CD43-mediated T-cell receptor microcluster formation to limit T cell activation [34], we propose that the augmented expression of ICAM1 and CD274 in CDCs may also contribute to CDCs’ immunosuppressive properties. 

Among all mRNAs analyzed, IP10/CXCL10 displayed the most pronounced change in expression compared with the control. This protein acts not only as a chemoattractant for monocytes and T-lymphocytes but also exhibits strong angiostatic properties. Specifically, it inhibits the migration of endothelial cells and the formation of vascular structures [35,36]. The marked increase in IP10 mRNA expression (up to 70,000-fold) and its concentration in the conditioned medium (reaching up to 12 ng/mL) (Figure 2b, Table 1) may explain why media from CDCs pre-incubated with cytokines and TLR ligands failed to amplify the pro-angiogenic response compared with untreated cells in a fibrin gel angiogenesis model (Figure 3d). This is notable, especially when considering the augmented expression and secretion of several pro-angiogenic factors such as VEGFA, MCP1, IL8, etc. (Figure 2b and Figure 3a,b).

We assessed not only the impact of the conditioned CDCs media on endothelial sprout growth but also the influence of the cells themselves. Notably, direct cell–cell interactions between CDCs and endothelial cells significantly promoted sprout outgrowth compared with the assay media alone (Appendix A). However, CDCs pre-incubation with the investigated factors did not exhibit a discernible influence on sprout development (Appendix A) (data is contained within Appendix A).

While our focus has primarily been on cytokine-induced changes in the secretion of growth factors and cytokines, it is essential to acknowledge the multifaceted nature of CDC secretions. Numerous studies have demonstrated that CDCs’ extracellular vesicles promote angiogenesis and reduce inflammation in cellular experiments and animal models [13,37,38].

In conclusion, our data suggest that both pro-inflammatory and anti-inflammatory stimulations exert a multifaceted influence on cardiosphere-derived cells, potentially altering their regenerative and immunomodulatory properties considerably.

## 3. Discussion

The vast landscape of regenerative medicine is continually evolving and embracing innovative methodologies and technologies. While newer methodologies, like those involving iPSCs, unquestionably hold promise, our study accentuates the versatility and adaptability of cardiosphere-derived cells. These cells have a robust ability to secrete factors that attenuate inflammation, bolster angiogenesis, and stimulate resident cells in the injured tissue by promoting a more favorable healing environment [27,28]. Our study underscores the potential of a long-established technique—cytokine pretreatment—as an avenue for enhancing the therapeutic efficiency of CDCs through the modulation of their secretion profile. While this approach has been extensively studied in another type of progenitor cells—mesenchymal stem cells (MSCs)—there is limited research concerning CDCs. Certain studies have indicated the beneficial effects of preconditioning CDCs with environmental stressors such as hypoxia, LPS, and H_2_O_2_ [18,19,39,40] as well as pharmacological agents like metformin [41]. Other approaches, including preconditioning with complex biological mixtures such as blood serum [42] and pericardial fluid post-myocardial infarction [12], have also demonstrated beneficial effects. However, studies on cytokines’ impact on cardiosphere-derived cells are limited, highlighting a significant research opportunity.

CDCs comprise a heterogeneous population of stromal vascular cardiac cells, which include cells resembling MSCs [6,7]. In our study, we partially confirmed this similarity by using antibodies targeting characteristic MSC antigens such as CD29, CD73, CD90, and CD105. However, we observed that CDCs express CD90 at a much lower level and tend to be slightly smaller in size than adipose-derived MSCs, as is evident in Figure 1a,b. Given the inherent MSC-like properties of certain CDC populations, it stands to reason that strategies effective for MSCs might be applicable to CDCs. Based on this idea, we explored cytokine pretreatment as a way to modify CDC behavior. We specifically selected cytokines that are expressed during different phases of cardiac inflammation. Notably, TNF-alpha and IFN-gamma are among the earliest cytokines to be upregulated, often within hours following the onset of cardiac injury or infection. These cytokines are pivotal in initiating the acute inflammation phase [43,44]. However, priming MSCs with a combination of TNF-alpha and IFN-gamma can increase the expression and secretion of immunosuppressive molecules such as indoleamine 2,3-dioxygenase (IDO), prostaglandin E2 (PGE2), and TGF-β [45]. This enhances the immunosuppressive potential of MSCs, suggesting an anti-inflammatory role of the TNF/IFN combination in that specific context. TGF-beta typically becomes prominent several days post-injury, persisting throughout the healing phase, with noted immunosuppressive and tissue-repairing properties. On the other hand, IL4 is more prevalent during the chronic phase, aiding in the resolution of inflammation and tissue remodeling. Furthermore, IL4 can reduce the expression of pro-inflammatory cytokines and chemokines [46]. Thus, these anti-inflammatory cytokines were chosen for their distinct roles at various stages of cardiac injury, offering a comprehensive insight into their potential influence on CDCs.

In addition to cytokines, we also tested TLR ligands, namely lipopolysaccharide and polyinosinic-polycytidylic acid. The concept of mesenchymal stem cell polarization into pro-inflammatory MSC1 and anti-inflammatory MSC2 phenotypes under the influence of these TLR ligands has been discussed in the literature. These MSC1 and MSC2 subtypes exhibit different functional attributes, notably in their impact on immune cells and in tissue repair [20,21]. However, the possibility of CDC polarization into analogous phenotypes such as “CDC1” and “CDC2” remains obscure. Moreover, studies on the effects of LPS on CDCs are very limited. Research conducted by Zhang and colleagues suggests that LPS pretreatment can significantly impact CDCs by enhancing their migratory capacity to ischemic areas, potentially through mechanisms involving β-catenin and HSP90 [40].

We hypothesized that CDCs might exhibit behavior similar to mesenchymal stem cells when exposed to TLR ligands and cytokines given the similarities in their response profiles and the ability of TLR ligands to influence cell fate decisions.

Our initial evaluation focused on how various cytokines affected CDC proliferation. To our surprise, none of the analyzed cytokines or TLR ligands had a notable impact on the viability of CDCs (Figure 1c), even though these cytokines are recognized for influencing the proliferation of various cell types. TLR activation can also influence cell proliferation. Both LPS and poly(I:C) have been shown in some studies to be capable of either promoting or suppressing cell proliferation based on the cell type and context [47]. Thus, the influence of TLR ligands and cytokines did not result in CDC death, but at the same time, it was insufficient to stimulate their proliferation. 

However, cytokines and TLR ligand treatment had a profound impact on the secretory profile of CDCs. Using a multiplex magnetic bead-based immunoassay, we assessed the concentration of 21 growth factors and cytokines in CDCs-conditioned media. We observed that the TLR3 ligand poly(I:C) did not exert a significant influence on the secretion of any of the analyzed factors (Figure 2b,c). 

Transforming growth factor (TGFb) led to an increase in the secretion of growth factors such as PDGFAA and VEGFA while reducing the secretion of EGF. The anti-inflammatory cytokine IL4 enhanced the secretion of eotaxin1, PDGFAA/AB, IL1a, FGF2, and IL15. Interferon-gamma stimulated the secretion of MCP3, IL15, IP10, and CCL5 while suppressing the secretion of IL6 and MCP1. However, the most substantial changes were observed with lipopolysaccharide (LPS) and the combination of TNF/INF. These factors primarily elevated the secretion of monocyte differentiation and attraction (MCP1, MCP3, G-CSF, GM-CSF, CCL5, IP10, MIP1a) and neutrophil recruitment cytokines (IL8, GRO1), thus potentially contributing to the mobilization of first-line defense cells during the acute phase of inflammation. 

Furthermore, we also observed a dramatic increase in the mRNA expression of the immunosuppressive factors CD274/PD-L1, IDO1/2, and ICAM1. Previous studies on MSCs have demonstrated the pivotal role of ICAM1 in MSC-mediated immunosuppression in direct co-cultures with lymphocytes [33]. Therefore, we additionally performed a flow cytometry analysis and confirmed an elevated expression of ICAM1 at the protein level, but only in response to TNF/INF stimulation of CDCs. This observation may explain why, of all treatments, only the TNF/INF combination suppressed PHA-stimulated lymphocyte proliferation more effectively than intact cells by about 10%. Therefore, despite the TNF/INF combination increasing the secretion of several strongly pro-inflammatory cytokines, it exhibits rather anti-inflammatory properties. This aligns with previous studies on MSCs, where the TNF/INF combination also demonstrated anti-inflammatory effects.

CDCs treated with LPS exhibit the weakest inhibition of lymphocyte proliferation; however, the secretion profile of LPS-treated CDCs closely resembles that induced by TNF/INF, preventing a definitive characterization of this phenotype as CDC1. 

Moreover, in all of the models that we used, poly(I:C) did not show significant deviations from the control, despite the close similarity of signaling pathways between TLR3 and TLR4. It is noteworthy that, upon treatment with LPS, CDCs display a significantly heightened secretion of a range of cytokines and chemokines, including G-CSF, GM-CSF, MCP3, IP10, MIP1a, RANTES/CCL5, and PDGFAA. This response markedly surpasses that seen in MSCs, suggesting a unique and robust activation profile in response to TLR4 signaling.

Conversely, when examining the response to TLR3 activation by the ligand poly(I:C), we did not observe a similar disparity in effect, as illustrated in the Appendix A. Future works should aim to elucidate the mechanisms behind these distinct responses. This observation raises important questions about the potential of CDCs to be polarized into distinct phenotypes similar to MSCs under the influence of TLR ligands. Consequently, we believe that additional studies are needed to definitively confirm the existence of two distinct CDC phenotypes that are akin to those documented for MSCs.

IL4 and TGF-beta-treated CDCs, in contrast to TNF/INF, primarily deviate by enhancing the secretion of PDGFAA. Notably, PDGF-A mRNA in infarcted myocardium significantly increased from week 2 onwards, while PDGF-B and PDGF-C expression declined, suggesting PDGF-AA’s role in cardiac repair and in maintaining cardiac connective tissue [30]. We hypothesize that modulating the CDC secretome using anti-inflammatory cytokines might promote the regenerative phase of cardiac repair.

It is worth noting that both TNF/INF and LPS induced an increase in the secretion of VEGFA by CDCs. VEGFA is a primary mediator of reparative angiogenesis, which begins during the acute inflammation phase and persists throughout the tissue repair period. To evaluate the pro-angiogenic potential of CDCs’ secretome, given the significance of VEGFA, we employed an in vitro angiogenesis model that involved embedding endothelial cell (EC)-coated microcarrier beads within a fibrin gel. However, conditioned media from the treated cells did not enhance endothelial sprout outgrowth when compared with untreated cells. Notably, media from CDCs treated with IL4 or TGF-beta even displayed a tendency to reduce sprout length. It is noteworthy that angiogenesis was assessed in the presence of complete endothelial growth medium (EGM2), which contains specific growth factors. 

In another experimental setup where CDCs were mixed in the gel together with EC-coated beads, facilitating direct cell–cell interactions and factor secretion in close proximity; the results were analogous. Adding non-stimulated cells significantly enhanced sprout outgrowth, possibly by direct cell contact establishment or by the release of exosomes; however, pre-incubation of CDCs with cytokines or TLR ligands did not stimulate this effect further. We hypothesize that this can be attributed to the fact that CDCs-conditioned media after treatment with INF, TNF/INF, and LPS contain a substantial amount of IP10 (CXCL10) (ranging from 5 to 12 ng/mL), markedly exceeding the concentration of VEGFA (ranging from 0.13 to 2 ng/mL). IP10 is known to exhibit potent angiostatic properties, inhibiting endothelial cell migration and the formation of vascular structures [35,36]. However, this cytokine also possesses anti-fibrotic properties and, as such, may function as an angiostatic and antifibrotic chemokine, delaying premature neovascularization and fibrosis until the damaged myocardium is cleared of apoptotic and necrotic cells [48]. 

Thus, we can speculate that CDCs beyond their role in regular cell turnover have a major role in the control of tissue inflammation and hold inherent potential in maneuvering the inflammatory environment by modulating the presence of various cytokines and chemokines.

In this study, we have explored the impact of cytokine pretreatment on the secretome of cardiosphere-derived cells (CDCs) and its potential implications for future clinical therapeutic strategies. While our focus has primarily been on cytokine-induced changes in the secretion of growth factors and cytokines, it is essential to acknowledge the multifaceted nature of CDC secretions and their potential relevance to cardiac regenerative medicine. Additionally, it is essential to recognize that CDCs secrete not only cytokines but also other bioactive factors, including exosomes and diuretic peptides. These secretions constitute a complex milieu that can influence the cardiac microenvironment. For instance, B-type natriuretic peptide, a diuretic peptide secreted by cardiac precursor cells, has been reported to possess remarkable anti-inflammatory and anti-fibrotic properties [49,50]. It is conceivable that the collective impact of various secretions, including cytokines, exosomes, and diuretic peptides, may contribute to the overall therapeutic effect of CDCs.

## 4. Materials and Methods

### 4.1. Cell Culture

Human cardiosphere-derived cells were obtained from the auricle of the right atrium during aortocoronary bypass surgery performed at the Department of Cardiovascular Surgery of the Federal State Budgetary Institution National Medical Research Centre of Cardiology named after academician E.I. Chazov of the Ministry of Health of the Russian Federation, which was performed with the patient’s written consent and approved by the Ethical Committee of the Centre (permit #271 (27 September 2021)). The CDCs culture protocol was developed based on the method described for the TICAP clinical study [24]. The right atrial auricle tissue was minced into 1–2 mm^3^ pieces washed in phosphate-buffered saline (PBS) and treated with a collagenase A solution (2 mg/mL, Roche-Merck KGaA, Darmstadt, Germany). Cells were cultured on fibronectin-coated dishes (Imtek, Moscow, Russia) in DMEM-F12 supplemented with 10% FBS and ITS (Gibco-ThermoFisher Scientific, Waltham, MA, USA). Cells migrating spontaneously from the explants were detached with trypsin and seeded in low-adhesion conditions (poly-HEMA coated dishes, Merck KGaA, Darmstadt, Germany) in DMEM-F12/10% FBS/ITS supplemented with 40 ng/mL of bFGF and 20 ng/mL of EGF, forming a 3D culture. After 48 h, the fully formed spheroids were collected and transferred to fibronectin-coated dishes. The spheroids attached, and the cells spread out. The 2D culture was grown to the required density in DMEM-F12 supplemented with 10% FBS and ITS and used for experiments up to the third passage. 

#### 4.1.1. Isolation of Peripheral Blood Mononuclear Cells (PBMCs)

Peripheral blood lymphocytes were isolated from healthy volunteer donors using a standard protocol. Briefly, anticoagulant-treated whole blood was diluted at a 1:3 ratio with PBS and layered onto a Ficoll solution (density 1.077 g/mL) (PanEco, Moscow, Russia). Following centrifugation at 400× *g* for 30 to 40 min at 20 °C with the brake off, mononuclear cells at the density gradient interface layer were gently collected. Then, cells were washed twice, seeded on 24-multiwell plates, and then labeled with 5 µM CMFDA (ThermoFisher Scientific, Waltham, MA, USA) for 30 min at 37 °C in a 5% CO_2_ atmosphere.

#### 4.1.2. Coculture of PBMCs with Cardiosphere-Derived Cells (CDCs)

For immunomodulation assays, CDCs were seeded in 24-multiwell plates at a density of 50 × 10^3^ cells/well. Cells were cultured in X-VIVO 15 serum-free media (Lonza, Bazel, Switzerland) and stimulated with investigated factors. The next day, the medium was aspirated and replaced with CMFDA-labeled PBMCs, which were simultaneously activated with phytohaemagglutinin-L (PanEco, Moscow, Russia) in X-VIVO 15 media. Cells were cocultured at a ratio of 1:10 (CDCs to MNC) for 3 days. Subsequently, non-adherent cells were harvested and analyzed using an FACSAria II flow cytometer (BD Bioscience, San Jose, CA, USA). Data were analyzed using FCS Express 7 Research Edition (De Novo Software, Pasadena, CA, USA).

### 4.2. Cytokine Treatments

TLR Agonists: Bacterial LPS (10 and 100 ng/mL) and polyinosinic-polycytidylic acid (p(I:C), 1 μg/mL, Sigma-Aldrich-Merck KGaA, Darmstadt, Germany) were added for 1 h. Subsequently, the cells were washed three times with culture medium, and conditioned media were collected 48 h post-induction.

Cytokines: TGF-beta 1 (2 and 20 ng/mL), IFNg (20 and 100 ng/mL), IL4 (25 ng/mL), and a combination of TNFα and IFNg (100 ng/mL) (R&D Systems, Minneapolis, MN, USA) were used. Cells were incubated with these cytokines overnight. Following this, cells were washed three times with the cultivation medium, and conditioned media were collected after 24 h. After collection, all media were cleared by centrifugation and stored at −70 °C until analysis. 

### 4.3. Cell Viability Assay

To study the viability, cells were seeded onto 3 separate 96-well plates (Corning Inc., New York, NY, USA) at a density of 5000 cells per well. Additionally, serial dilutions of cells were prepared to establish a calibration curve. Five hours post-seeding, once complete cell adhesion was achieved, the above-specified factors were introduced at the indicated concentrations. Every 24 h, 10 μL of the resazurin-based Presto Blue^TM^ dye (Life Technologies—ThermoFisher Scientific, Waltham, MA, USA) was added to the appropriate wells. This dye undergoes a fluorescence spectrum shift from blue to red upon reduction by intracellular enzymes. An hour after the dye’s addition, the fluorescence intensity was measured using a Victor3 plate reader (Perkin Elmer, Waltham, MA, USA) at a wavelength of 600 nm. Using the calibration curve established from the serially diluted cells, the changes in cell numbers over time in response to the polarizing factors were determined.

### 4.4. Flow Cytometry 

Cardiosphere-derived cells (CDCs) were harvested from culture dishes washed with PBS and then incubated with fluorescent primary antibodies or isotype control IgG in 1% BSA-PBS for 30 min at 4 °C, as per manufacturer’s guidelines. The antibodies used targeted mesenchymal markers (CD105, CD73, CD90), endothelial marker CD31, and ICAM1 (all—BD Bioscience, San Jose, CA, USA). When unlabeled antibodies against PAI2 (ThermoFisher Scientific, Waltham, MA, USA) and CD29 (Abcam, Cambridge, UK) were employed, a subsequent incubation with fluorescent secondary antibodies (Thermo Fisher Scientific Inc.,Waltham, MA, USA) was performed for 1 h. PAI2 staining required cell permeabilization using 0.2% saponin, and all related steps were performed in 1% BSA-PBS with 0.2% saponin. After staining, cells were either immediately analyzed or fixed in 1% formalin for later analysis using a FACSCanto™ II cytometer (BD Bioscience, San Jose, CA, USA). A total of 1 × 10^4^ events were recorded for each sample. Data were analyzed using FCS Express 7 Research Edition (De Novo Software, Pasadena, CA USA).

### 4.5. qRT-PCR Analysis

Total RNA was extracted using the RNeasy Mini Kit (QIAGEN, Germantown, MD, USA), and 1 µg of the total RNA for each sample was reverse-transcribed using the High Capacity cDNA Reverse Transcription Kit (ThermoFisher Scientific, Waltham, MA, USA) following the manufacturer’s protocol. A qRT-PCR analysis was performed on a StepOnePlus™ Real-Time PCR System (Applied Biosystems/ThermoFisher Scientific, Waltham, MA, USA). The reaction mixture contained an SYBR Green PCR Master Mix (Syntol, Moscow, Russia), specific primers (10 pmol each), and 1–5 ng of cDNA. The program was set to 95 °C for 10 min followed by 40 cycles of 95 °C for 15 s and 60 °C for 60 s. A melt curve analysis was performed post-amplification. Primer sequences (Eurogen, Moscow, Russia) for genes were: IL6 (for: ACT CAC CTC TTC AGA ACG AAT TG; rev: CCA TCT GAA GGT TCA GGT TG); IL1β (for: ACA GAT GAA GTG CTC CTT CCA; rev: GTC GGA GAT TCT AGC TGG AT); VEGFa (for: TGT GTG TGT GTG AGT GGT TGA; rev: TCT CTG TGC CTC GGG AAG); GAPDH TGC ACC ACC AAC TGC TTA; rev: GGC ATG GAC TGT GGT CAT GAG); Cd274 (for: TGG CAT TTG CTG AAC GCA TTT; rev: TGC AGC CAG GTC TAA TTG TTT T); HLA-DRB1 (for: TGC CAA GTG GAG CAC CCA A; rev: GCA TCT TGC TCT GTG CAG AT); CXCL1/GRO1 (for: AAC CGA AGT CAT AGC CAC AC; rev: CCT CCC TTC TGG TCA GTT G);CCL2/MCP1 (for: AGT CTC TGC CGC CCT TCT; rev: GTG ACT GGG GCA TTG ATT G); GM-CSF = CSF2 (for: TCT CAG AAA TGT TTG ACC TCC A; rev: GCC CTT GAG CTT GGT GAG); A); MCP3/ccl7 (for: ACA GAA GGA CCA CCA GTA GCC A; rev: GGT GCT TCA TAA AGT CCT GGA CC); CSF3/G-CSF (for: GCT GCT TGA GCC AAC TCC ATA; rev: GAA CGC GGT ACG ACA CCT C); eotaxin/CCL11 (for: CCC CTT CAG CGA CTA GAG AG; rev: TCT TGG GGT CGG CAC AGA T); RANTES/CCL5 (for: CGC TGT CAT CCT CAT TGC TA; rev: GCA CTT GCC ACT GGT GTA GA); STAT1 (for: ATG GCA GTC TGG CGG CTG AAT T; rev: CCA AAC CAG GCT GGC ACA ATT G); SERPINEB2/ PAI2 (for: CAT GGA GCA TCT CGT CCA C; rev: ACT GCA TTG GCT CCC ACT T); CXCL10/IP10 (for: GTG GCA TTC AAG GAG TAC CTC; rev: TGA TGG CCT TCG ATT CTG GAT T).

Gene expression levels were normalized to beta-actin (bActB) and glycerylalde-hyde-3-phosphate dehydrogenase (GAPDH) using the comparative 2^−ΔΔCt^ method.

### 4.6. Multiplex Immunoassay (Magpix)

To study the potential modulation of cytokine secretion profiles, cells were exposed to the following factors: LPS (10 and 100 ng/mL), p(I:C) (1 µg/mL), TGFb1 (2 and 20 ng/mL), IFNg (20 and 100 ng/mL), IL4 (25 ng/mL), and a combination of TNFα and IFNg (each at 20 ng/mL) at the specified concentrations. After 18 h of incubation, the cells were washed thrice with culture medium; conditioned media were subsequently collected 24 h post-treatment.

The cytokines quantification in the conditioned media was performed using the MILLIPLEX MAP 41 Human Cytokine/Chemokine Magnetic Bead Panel (HCYTMAG-60K-PX4141, Merck, Darmstadt, Germany) in strict adherence to the manufacturer’s guidelines. Initially, protein standards were serially diluted. Then, 25 µL aliquots of each standard dilution, control mixes (supplied by the manufacturer), and conditioned media samples were pipetted into a 96-well assay plate in duplicate. This was followed by the addition of fluorescent magnetic beads conjugated with cytokine-specific capture antibodies into each well. The plate was sealed and subjected to an overnight incubation at 4 °C with gentle shaking. Post-incubation, the plate was washed three times to remove unbound substances. Subsequently, a biotinylated detection antibody was added to each well and incubated for 1 h at room temperature. After a series of washes, streptavidin phycoerythrin was added, and the plate was further incubated for 30 min. Fluorescence detection and quantification were conducted using the MAGPIX^®^ System (Luminex-Merck, Darmstadt, Germany). The acquired data were analyzed utilizing xPONENT software 4.3.229.0, and cytokine concentrations in the conditioned media were extrapolated from the standard curves.

### 4.7. Fibrin Gel Angiogenesis Bead Assay

Endothelial cells (HUVECs) were gifted by Dr. Olga Antonova (Cell Adhesion department, National Medical Research Center for Cardiology). Cells were cultured in EGM-2 medium supplemented with angiogenic factors. For assays, cells were detached using trypsin-EDTA seeded on dextran Cytodex beads (Sigma-Aldrich-Merck, Darmstadt Germany) overnight and subsequently labeled with 5 µM CMFDA fluorescent cell tracker (ThermoFisher, Waltham, MA, USA) for 30 min at 37 °C in a 5% CO_2_ atmosphere.

A fibrin gel matrix was then prepared by mixing an HUVEC-coated beads suspension with fibrinogen solution (2.5 mg/mL). Thrombin (0.5 U/mL final concentration) was then added to initiate gelation. This gel was immediately dispensed into 12-well chamber slides (Ibidi, Gräfelfing, Germany) (200 µL/well) and allowed to polymerize at 37 °C for 1 h. After polymerization, EGM-2 culture medium alone or mixed with conditioned media from CDCs (200 µL EGM2 + 50µL CM) was added, and the cells were incubated for 5 days at 37 °C in 5% CO_2_. The length of endothelial sprouts was measured from the acquired photomicrographs using ImageJ/Fiji software version 1.53t (NIH, Bethesda, MD, USA).

### 4.8. Statistical Analysis

Differences between experimental groups were assessed by Student’s *t*-test. The values are expressed as the mean ± SD. *p* ≤ 0.05 was considered as statistically significant.

## 5. Conclusions

Together, our research underscores that TNF/INF treatment in CDCs significantly enhances the secretion of pro-inflammatory cytokines while modulating the expression of pivotal immunosuppressive factors. Though the effects of IL4 and TGF-beta on CDC secretion might seem less pronounced, they notably enhance PDG-FAA secretion in contrast to other factors tested, indicating their potential role in cardiac repair. This differential impact of cytokines on the CDC secretome illustrates the multifaceted roles of CDCs in cardiac regeneration, synergizing with immune cells to orchestrate cardiac repair. The marked presence of IP10, well known for its angiostatic attributes, presents an intriguing challenge. Further investigations are needed to confirm its possible anti-fibrotic capabilities and further delineate CDCs’ involvement in cardiac fibrosis. Collectively, our study elucidates how various cytokine treatments shape CDC secretomes, providing invaluable insights for future therapeutic interventions in cardiac repair.

### Study Limitations

It is important to note that we did not investigate the effects of cytokine pretreatment on CDC efficacy in vivo. This was because our focus was on studying cytokine secretion, and the observed changes in the secretome of human cells may not be directly relevant to mouse models. For instance, the effects of IP10 on angiogenesis vary between humans and mice due to differences in the isoforms of its receptor, CXCR3. Its second isoform, known as CXCR3-B, has been identified in humans but not in mice. CXCR3-B is primarily expressed by endothelial cells and mediates IP10 antiangiogenic properties by promoting apoptosis and inhibiting endothelial cell proliferation and migration [48]. Moreover, there are other significant differences. Mice lack a direct analogue of IL-8 and its receptor CXCR1 [51]. However, in our research, we observed that the concentrations of IL-8 and another potent neutrophil attractant, GRO-α (growth-related oncogene-α, CXCL1), which shares the CXCR2 receptor with IL-8, were among the highest in the CDC media in the intact control (ranging from 0.7 to 2 ng/mL) and increased after treatment (up to 11 ng/mL). Additionally, the balance of lymphocytes and neutrophils is quite different; human blood is neutrophil-rich (50–70% neutrophils, 30–50% lymphocytes) whereas mouse blood has a strong preponderance of lymphocytes (75–90% lymphocytes, 10–25% neutrophils) [51]. As a result, mouse models may not faithfully replicate the effects on the neutrophil component of immunity. These observations underscore the limitations of using mouse models to fully reproduce the effects on the neutrophilic arm of the immune system.

## Figures and Tables

**Figure 1 ijms-24-17278-f001:**
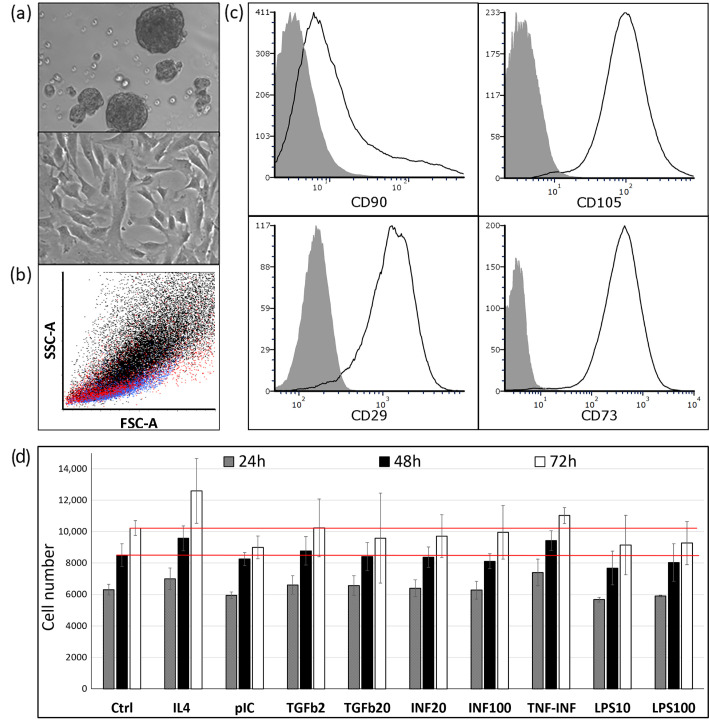
(**a**) Top—photomicrograph of cardiospheres on poly-HEMA-coated culture dish after 48 h of culture in growth factor-supplemented medium; middle—photomicrograph of cardiosphere-derived cells. Phase contrast images at 200× magnification; (**b**) Flow cytometric comparative analysis of MSC (black), CDCs (red), and HUVEC (blue) by size. (**c**) Representative flow cytometry histograms of CDCs. Passaged CDCs were stained with fluorescently labeled antibodies against CD29, CD73, CD90, and CD105. Control staining was performed using PE, PE-Cy7, and APC-conjugated isotype-matched IgG; (**d**) Effect of cytokines and TLR ligands on CDC proliferation. Concentrations used: pIC: 1 μg/mL; IL4: 25 ng/mL; TGFb: 2 ng/mL (TGFb2) and 20 ng/mL (TGFb20); INFg: 20 ng/mL (INF20) and 100 ng/mL (INF100); LPS: 10 ng/mL (LPS10) and 100 ng/mL (LPS100); TNF/INF: 20 ng/mL each of TNFa and INFg. “Ctrl” assay medium—DMEM-F12/ITS/0.5% FBS. Red lines indicate the average number of cells in the “Ctrl” group at 48 or 72 h. Data are presented as the mean ± standard deviation. Cell viability assessed via PrestoBlue^TM^ staining.

**Figure 2 ijms-24-17278-f002:**
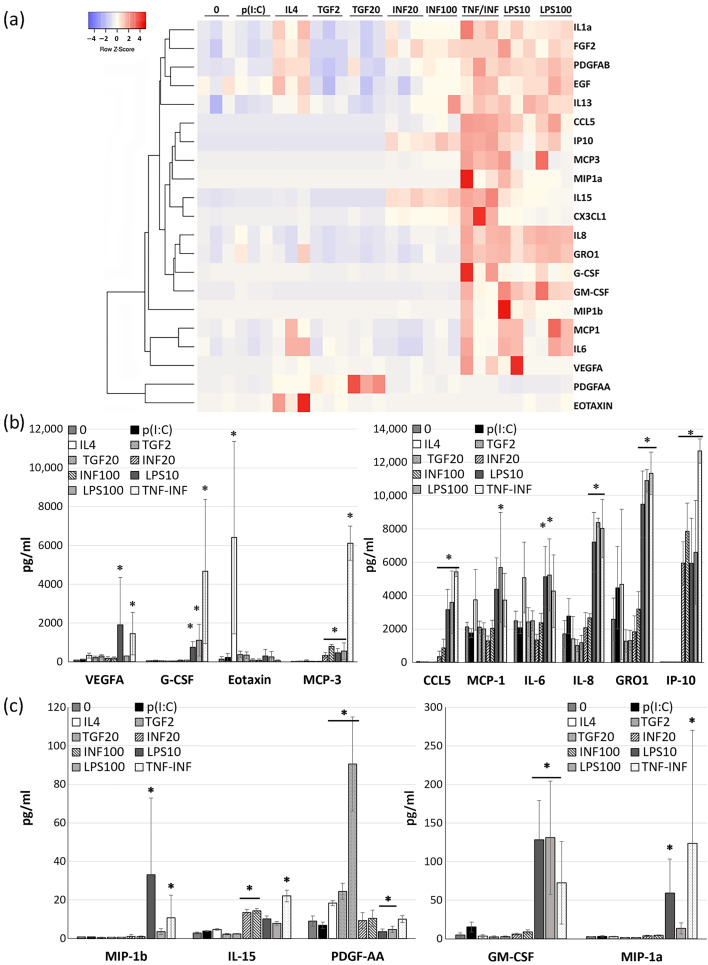
Results of a multiplex immunoassay (Magpix, HCYTMAG-60K-PX41) of the quantity of growth factors and cytokines in CDCs supernatants collected 24 h after cell incubation with TLR ligands for 1 h and cytokines for 18 h. (**a**) Heatmap illustrating changes in the secretory profile for each of the three donors. The clustering was performed using the average linkage method, with Kendall’s tau being used as the distance measurement metric. Each column represents the specific stimulation conditions, and each cell color intensity indicates the concentration (pg/mL), with the red color indicating higher values and the blue color indicating lower values. The dendrogram on the left visualizes the hierarchical clustering of the cytokines based on their similarity. (**b**,**c**) Histograms of cytokine concentrations (pg/mL) in CDCs-conditioned media influenced by studied factors. Data are presented as the mean ± SD; *n* = 3. Significance from media of intact (without any stimulation) cell (“0”): *—*p* < 0.05 (Student’s *t*-test).

**Figure 3 ijms-24-17278-f003:**
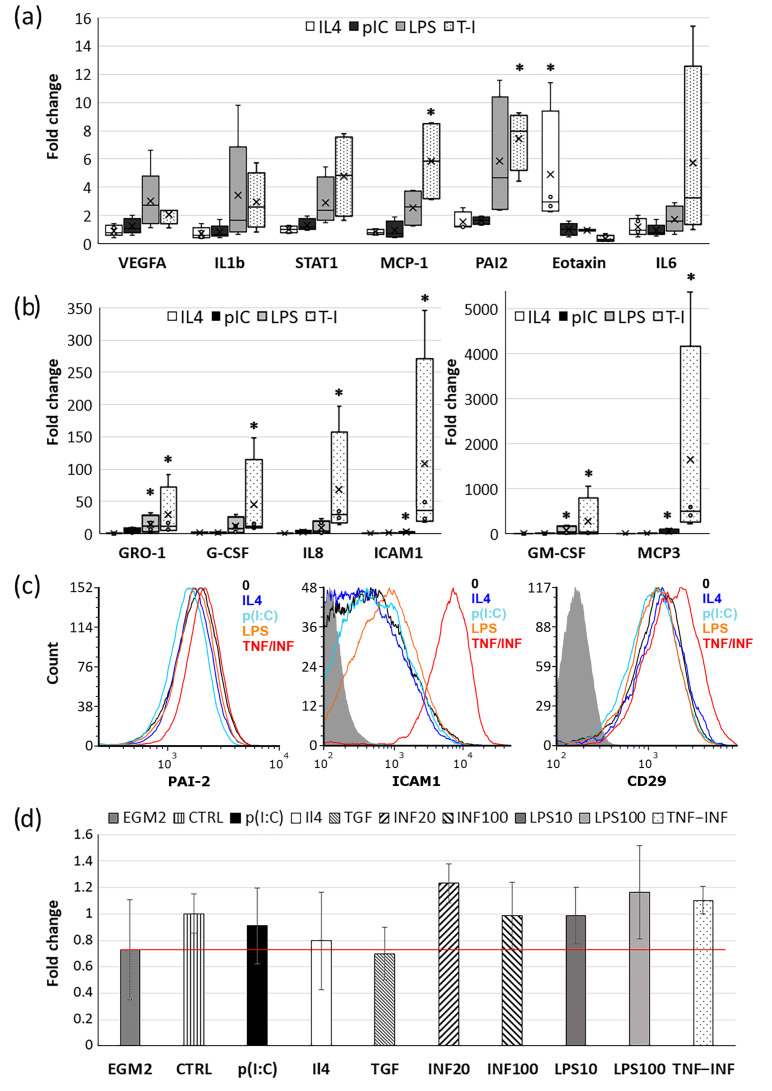
Effects of cytokines and TLR ligands on CDC gene expression. (**a**,**b**) Box plots of relative expression of pro- and anti-inflammatory genes in CDCs following incubation with IL4, p(I:C), LPS, and TNF/INF for 18 h. Expression changes are normalized to the mRNA levels of the intact cell (without any stimulation). Box edges mark the 25th and 75th percentiles. Whiskers extend to 1.5 times the interquartile range from the 1st to the 3rd quartile; n = 4. Significance from unstimulated cell: *—*p* < 0.05 (Student’s *t*-test). (**c**) Representative flow cytometry histograms displaying a typical distribution of fluorescence intensity of CD29, ICAM1, and PAI2 in CDCs after treatments. (**d**) Results from the fibrin gel angiogenesis bead assay. CDCs-conditioned media were mixed with endothelial growth media EGM2 (1:5) and added to HUVEC-coated Cytodex beads embedded in fibrin gel. Endothelial sprout lengths were quantified after 5 days of culturing. Graphs represent the fold changes relative to media from the intact cells (“CTRL”). Red line showed relative sprout legth in EGM2 without CDC media -”EGM2”. The red line shows the relative change in sprout length in beads treated with EGM2 without CDC medium, "EGM2". Data are presented as the mean values ± SD.

**Table 1 ijms-24-17278-t001:** Fold change in mRNA expression in CDCs under pro- and anti-inflammatory stimulation relative to intact cells.

Genes	IL4	p(I:C)	LPS (*)	TNF/INF (*)
*CD274*	0.68 ± 0.27	1.14 ± 0.32	2.67 ± 0.13	38.57 ± 30.51
*HLA-DRB1*	1.12 ± 0.28	2.29 ± 1.09	7.03 ± 2.73	699.55 ± 187.95
*IDO1/2*	1.31 ± 0.71	3.23 ± 1.89	35.05 ± 41.07	9741.64 ± 5122.44
*IP10*	2.48 ± 3.10	1.81 ± 2.44	315.23 ± 242.77	70,727.20 ± 18,141.92

Data are presented as the mean ± SD; *n* = 4. Significance relative to intact cells (without any stimulation) cell: *—*p* < 0.05 (Student’s t-test).

## Data Availability

Data is contained within the article and Appendix A.

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
