# Peer review of "Paracrine Responses of Cardiosphere-Derived Cells to Cytokines and TLR Ligands: A Comparative Analysis"

_ijms, 2023, doi:10.3390/ijms242417278_

Round 1

Reviewer 1 Report

Comments and Suggestions for Authors

Reviewing the manuscript entitled, “Paracrine Responses of Cardiosphere-Derived Cells to Cytokines and TLR Ligands: A Comparative Analysis.” by Zubkova E et al., this focuses on an evaluation of secretome changes of CDCs stimulation by cytokines and TLR ligand. Although this is an interesting manuscript, since this study only used CDC, comparisons with other MSCs and characteristics of CDCs cannot be seen. Considering the pathological condition in the heart, the authors should conduct a change from the acute stage to the chronic stage. The authors need to respond to the following concerns.

 In Figure 1, Figure order and figure legend order do not match. The author should edit it because it is difficult to read.

 In Figure 2, data showed CDCs responded to LPS but not to TLR ligands. Is this a phenomenon seen in other MSCs as well? Or is it CDCs specific? I think it is important to compare CDCs with other MSCs.

 In Figure 2, although the figure is too small to confirm, it seems that there is no dose dependence between LPS10 and LPS100 in some items.

 Although the authors mentioned “Transforming growth 170 factor (TGFb) augmented only the secretion of growth factors PDGFAA and VEGFA.” from line 170 to 171, Figure 2b is too small to confirm. The authors need to modify it.

 What kind of cardiovascular pathology is LPS stimulation used in CDCs?

 The authors described future clinical therapeutic method of CDCs in this manuscript. If so, is it important to have a cytokine stimulation time schedule that follows the clinical course? In addition, CDCs may secrete not only cytokines but also exosomes and diuretic peptides. For example, CNP has extremely high anti-inflammatory and anti-fibrotic properties. The authors should mention that there are also effects of these other secretions in the discussion section.

 The authors mentioned “This approach aims to optimize the therapeutic potential of CDCs in mitigating inflammatory and degenerative cardiac conditions. By shedding light on these processes, we aspire to optimize the therapeutic utility of CDCs, thereby opening new doors in the field of cardiac regenerative medicine.” However, there are no results of treatment efficacy, and this statement is a bit of an exaggeration. The authors need to edit it.

 The authors need to correct the error in the number.

Comments on the Quality of English Language

There are no major problems, but small mistakes are noticeable.

Author Response

We appreciate the reviewer's thoughtful analysis and constructive feedback on our manuscript. Below, we address each of the concerns raised:

Reviewer's Comment 1: "In Figure 1, Figure order and figure legend order do not match. The author should edit it because it is difficult to read."

Response: We apologize for this oversight and have revised Figure 1 to ensure that the figure order aligns correctly with the figure legend for improved readability.

Reviewer's Comment 2: "Data showed CDCs responded to LPS but not to TLR ligands. Is this a phenomenon seen in other MSCs as well? Or is it CDCs specific? I think it is important to compare CDCs with other MSCs."

Response: We acknowledge the importance of this comparison. We have added a discussion on how CDCs' response to LPS and TLR ligands might be similar or different from other MSCs, and we have expanded our supplementary section to present results on the secretome changes in MSCs under the influence of LPS and poly(I:C) to provide a more comprehensive analysis. All changes in the text for convenience are highlighted in yellow.

Results, line 198: “Our initial assumption was that MSCs and CDCs would exhibit similar responses in secretion changes following TLR-ligand stimulation. However, it turned out that CDCs significantly increase the secretion of a range of factors in response to LPS, namely G-CSF, GM-CSF, MCP3, IP10, MIP1a, RANTES/CCL5, and PDGFAA. In the case of the TLR3 ligand poly(I:C), no significant difference in impact was noted (Supplementary Figure 3).”

Discussion line 401: “It is noteworthy that, upon treatment with LPS, CDCs display a significantly heightened secretion of a range of cytokines and chemokines, including G-CSF, GM-CSF, MCP3, IP10, MIP1a, RANTES/CCL5, and PDGFAA. This response markedly surpasses that seen in MSCs, suggesting a unique and robust activation profile in response to TLR4 signaling.

Conversely, when examining the response to TLR3 activation by the ligand poly (I:C), we did not observe a similar disparity in effect, as illustrated in Supplementary Figure 3. Future work should aim to elucidate the mechanisms behind these distinct responses. This observation raises important questions about the potential of CDCs to be polarized into distinct phenotypes similar to MSCs under the influence of TLR ligands. Consequently, we believe that additional studies are needed to definitively confirm the existence of two distinct CDC phenotypes akin to those documented for MSCs.”

Reviewer's Comment 3: "Although the figure is too small to confirm, it seems that there is no dose dependence between LPS10 and LPS100 in some items."

Response: Thank you for your observation. We hope that In the Web version of the article, the figures can be significantly enlarged, which should aid in better data visualization. We have carefully re-examined our data and confirm that we did not find any significant differences between the LPS doses. We included this sentence to paper, line 173 “It is important to note that no significant differences in response were detected across the administered doses, whether for TGF, INF, or LPS treatments.”

Reviewer's Comment 4: "Although the authors mentioned 'Transforming growth factor (TGFb) augmented only the secretion of growth factors PDGFAA and VEGFA.' from line 170 to 171, Figure 2b is too small to confirm."

Response: To examine these data in more detail, we have modified the presentation of Figure 2b by dividing it into two parts. This allowed us to increase the scale size, thus improving the visibility of the data. The updated version of the figure is included in our revised manuscript.

We hope that this improvement will make the analysis of the results more visually informative and beneficial for readers and reviewers.

Reviewer's Comment 5: "What kind of cardiovascular pathology is LPS stimulation used in CDCs?"

Response: Lipopolysaccharide (LPS) is known to trigger cardiac dysfunction and heart failure, particularly in the context of sepsis. Sepsis-induced cardiomyopathy is a severe multiorgan injury that includes cardiac contractile dysfunction, which can manifest as dilated heart chambers, impaired contractility, and reduced cardiac output. LPS from gram-negative bacteria plays a pivotal role in this condition, leading to cardiomyocyte apoptosis due to excess inflammation and the generation of reactive oxygen and nitrogen species, causing tissue damage in the heart. Moreover, LPS from gram-negative bacteria participate in  bacterial myocarditis.

However, inflammatory processes have dual roles in cardiac injury and repair, with LPS inducing dysfunction in septic cardiomyopathy while its controlled pretreatment might protect against ischemia/reperfusion injury. The influence of LPS on MSCs is well-documented, enhancing their reparative abilities and myocardial regeneration.

Some studies have suggested that an appropriate concentration of LPS pretreatment of MSC may protect myocardial cells from apoptosis induced by ischemia/reperfusion injury (Ha et al., 2008; Yao YWet.al., 2009). It has been reported that LPS preconditioning of MSCs upregulates VEGF and activates PI3K/Akt axis, which led to improved survival and angiogenesis induction in the animal model of acute myocardial infarctions (Yao YW,2010). However, focused studies on LPS's effects on CDCs are rarer but we could suggest beneficial effects similar to MSC, partially this supported by Zhang et al.'s work. They showed that  LPS pretreatment can significantly impact CSCs, enhancing their migratory capacity to ischemic areas, potentially through mechanisms involving β-catenin and HSP90. This indicates a similar therapeutic promise for CDCs in ischemic heart conditions, meriting further exploration

We have expanded discussion to detail the known influence of LPS stimulation in CDCs, providing more context to our findings.

Line 354 “Moreover, studies on the effects of LPS on CDCs are very limited. Research conducted by Zhang and colleagues suggests that LPS pretreatment can significantly impact CDCs by enhancing their migratory capacity to ischemic areas, potentially through mechanisms involving β-catenin and HSP90”.

Reviewer's Comment 6: " The authors described future clinical therapeutic method of CDCs in this manuscript. If so, is it important to have a cytokine stimulation time schedule that follows the clinical course? In addition, CDCs may secrete not only cytokines but also exosomes and diuretic peptides. For example, CNP has extremely high anti-inflammatory and anti-fibrotic properties. The authors should mention that there are also effects of these other secretions in the discussion section."

Response:

While the timing of cytokine treatment may play a role in optimizing the therapeutic potential of CDCs, aligning cytokine pretreatment with the clinical course is not standard practice. Typically, published protocols suggest a short-term stimulation, usually over one to two days.

We have included a short discussion on the role of cytokines, exosomes, and diuretic peptides, but unfortunately, we did not find any articles on the secretion of CNP by cardiosphere cells. We only managed to find work on the secretion of BNP.

Line 446. “In this study, we have explored the impact of cytokine pretreatment on the secretome of cardiosphere-derived cells (CDCs) and its potential implications for future clinical therapeutic strategies. While our focus has primarily been on cytokine-induced changes in the secretion of growth factors and cytokines, it is essential to acknowledge the multifaceted nature of CDC secretions and their potential relevance to cardiac regenerative medicine. Additionally, it is essential to recognize that CDCs secrete not only cytokines but also other bioactive factors, including exosomes and diuretic peptides. These secretions constitute a complex milieu that can influence the cardiac microenvironment. For instance, B-type natriuretic peptide is diuretic peptide secreted by cardiac precusor cells, has been reported to possess remarkable anti-inflammatory and anti-fibrotic properties. It is conceivable that the collective impact of various secretions, including cytokines, exosomes and diuretic peptides, may contribute to the overall therapeutic effect of CDCs.”

Reviewer's Comment 7: " The authors mentioned “This approach aims to optimize the therapeutic potential of CDCs in mitigating inflammatory and degenerative cardiac conditions. By shedding light on these processes, we aspire to optimize the therapeutic utility of CDCs, thereby opening new doors in the field of cardiac regenerative medicine.” However, there are no results of treatment efficacy, and this statement is a bit of an exaggeration. The authors need to edit it.

Response: We have revised this statement to more accurately reflect the current findings and have included a caveat about the preliminary nature of our results concerning treatment efficacy.

The revised text (line 88) now reads: “Our study aims to deepen our understanding of how cytokine pretreatment can modulate the secretome of CDCs, which may, in turn, enhance their therapeutic potential. Our findings lay the groundwork for future in vivo studies that will directly assess the efficacy of such pretreatment strategies in the context of cardiac regenerative medicine.”

We appreciate this opportunity to clarify our objectives and have adjusted the manuscript accordingly to avoid any overstatement regarding the direct treatment efficacy in the current study.

Reviewer's Comment 8: "The authors need to correct the error in the number."

Response: We have carefully reviewed the manuscript and corrected the numerical error.

We hope that these revisions and clarifications address the reviewer's concerns effectively and enhance the quality and impact of our manuscript. We are grateful for the opportunity to improve our work through this valuable feedback.

Reviewer 2 Report

Comments and Suggestions for Authors

This is a nice manuscript with a clear question and straight forward methodology. However, I have some remarks that should be addressed:

-        Ll. 51 – 64. Some information is doubled. The authors should improve the reading quality by rewriting this paragraph.

-        Introduction: Authors should describe in more detail and with more literature background why they think that preconditioning of CDCs could be beneficial.

-        Axis labeling is missing in nearly every graph

-        Ll. 135 – 137: The authors should explain which cytokines and TLR-ligands were used and why they decided to use especially these ones.

-        L. 210: what is meant with intact cell?

-        Ll. 274 – 278: There are other studies investigating the influence of preconditioning cardiac progenitor cells with biological stimuli (for example doi: 10.3390/cells9061472, DOI: 10.1161/CIRCRESAHA.109.197723). This should be noticed in the discussion.

-        Given the high complexity of pro-inflammatory and anti-inflammatory stimulation on CDCs, the authors are encouraged to provide a summarizing figure as conclusion depicting potential direct and paracrine effects of CDCs. This would greatly enhance the understanding for the reader.

-        Ll. 259 – 263 + ll. 358 – 383: in the context of direct cell-cell interactions vs. paracrine effects in angiogenesis, it should be noted that extracellular vesicles of cardiac progenitors indeed are able to act anti-inflammatory and proangiogenic in other studies (for example https://doi.org/10.1093/eurheartj/ehad636). This should be discussed.

Author Response

Thank you for your insightful comments and careful evaluation of our manuscript. We appreciate your dedication to improving the quality of the work presented. We are confident that these amendments will contribute to a clearer and more impactful paper.

Reviewer's Comment 1: " Ll. 51 – 64. Some information is doubled. The authors should improve the reading quality by rewriting this paragraph.”

Response: Thank you for your insightful comments and for drawing attention to the redundancy in lines 51-64. We have thoroughly reviewed and revised the mentioned paragraph to eliminate any repetition and enhance the clarity of the manuscript. The revised text is presented below and has been highlighted in yellow in the manuscript for ease of review.

Line 59 “A significant role in the regulation of CDC secretome is played by inflammation-related signals [11, 12, 13]. During the acute phase of myocardial infarction, a pro-inflammatory environment is formed under the influence of free radicals and cytokines [14]. This environment serves dual purposes: activating and accumulating cells for debris resorption and preparing for the proliferative healing phase. As such, post-infarct heart repair involves a transition from a pro- to an anti-inflammatory environment, modulated by various mediators (cytokines, growth factors, exosomes/microvesicles, microRNAs) affecting all cell types involved in heart damage and repair”

Reviewer's Comment 2: “Introduction: Authors should describe in more detail and with more literature background why they think that preconditioning of CDCs could be beneficial.”

Response: Thank you for your valuable input. We've expanded the Introduction to provide a clearer explanation of CDC preconditioning, including its benefits, and we've enriched this section with pertinent literature. These revisions aim to directly address your request for more detail and background.

Line 61: “While the application of stromal cells for tissue regeneration is a promising approach, it encounters several significant challenges. Factors such as oxidative stress, acidosis, hypoxia, inflammation, and the inadequate supply of nutrients and oxygen in the injured area create a hostile microenvironment, hindering the efficient participation of stromal cells in the regeneration process. To address these challenges, a strategy known as 'cell preconditioning' were proposed. This approach serves a dual purpose: firstly, it amplifies specific beneficial cell signaling pathways, and secondly, it exposes cells to stress conditions resembling the harsh environment they will encounter in the injured area. The goal of this method is to prepare cells to adapt effectively to the hostile host environment before administration.

Extensive research into the preconditioning of MSCs, including exposure to hypoxia or the use of cytokines, pharmacological and chemical agents has been performed to enhance the overall efficacy of MSC transplantation. All these techniques promote MSC survival and their capacity to form a regenerative and proliferative environment. However, comprehensive investigations into the preconditioning of CDCs remain relatively limited, although certain studies have indicated the beneficial effects of preconditioning CDCs with hypoxia and H2O2.”

Reviewer's Comment 3: “Axis labeling is missing in nearly every graph”.

Response: Thank you for bringing this to our attention. We have now added axis labels to nearly all figures where it was feasible to enhance the clarity and readability of the graphs.

Reviewer's Comment 4: “Ll. 135 – 137: The authors should explain which cytokines and TLR-ligands were used and why they decided to use especially these ones.”

Response: Thank you for your question. We have updated our manuscript to describe the rationale for selecting specific cytokines and TLR-ligands for our work. The revised text can be found below. We hope this adequately addresses your query and strengthens the manuscript.

Lines 154- 162. “We specifically selected cytokines that are expressed during different phases of cardiac inflammation: TNF-alpha and IFN-gamma are among the earliest cytokines to be upregulated. TGF-beta typically becomes prominent several days post-injury, persisting throughout the healing phase, and IL-4 is more prevalent during the chronic phase, aiding in the resolution of inflammation and tissue remodeling. In accordance with the concept proposed by the Betancourt group, TLR3 ligand poly(I:C) has the potential to induce an anti-inflammatory phenotype, while TLR4 ligand LPS may induce a pro-inflammatory phenotype in MSCs. However, it's important to note that these ligands actions in the context of CDCs have not been extensively explored.” 

Reviewer's Comment 5: “L. 210: what is meant with intact cell?”

Response: By 'intact', we refer to cells that have not undergone any stimulation or manipulation. We have clarified this in the manuscript by adding an explanation in the figure descriptions where this term is used.

Reviewer's Comment 6: “Ll. 274 – 278: There are other studies investigating the influence of preconditioning cardiac progenitor cells with biological stimuli (for example doi: 10.3390/cells9061472, DOI: 10.1161/CIRCRESAHA.109.197723). This should be noticed in the discussion.”

Response: Thank you for highlighting additional studies on the influence of preconditioning cardiac progenitor cells with biological stimuli. We have acknowledged these studies in our discussion. Specifically, we included the articles you mentioned (doi: 10.3390/cells9061472, DOI: 10.1161/CIRCRESAHA.109.197723) and expanded on various preconditioning approaches in lines 317-324. This section now provides a more comprehensive overview of different preconditioning methods.

Line 317-324 “Certain studies have indicated the beneficial effects of preconditioning CDCs with environmental stressors such as hypoxia, LPS and H2O2 [18, 19], [39] [41] as well as pharmacological agents like metformin [42]. Other approaches, including preconditioning with complex biological mixtures, such as blood serum [40] and pericardial fluid post-myocardial infarction [12], have also demonstrated beneficial effects. However, studies on cytokine impact on cardiosphere-derived cells are limited, highlighting a significant research opportunity.”

Reviewer's Comment 7: “Given the high complexity of pro-inflammatory and anti-inflammatory stimulation on CDCs, the authors are encouraged to provide a summarizing figure as conclusion depicting potential direct and paracrine effects of CDCs. This would greatly enhance the understanding for the reader.”

Response: Thank you for the suggestion to provide a summarizing figure that illustrates the complex interactions of pro-inflammatory and anti-inflammatory stimulation on CDCs. We have created a graphical abstract for our paper, which succinctly represents the potential direct and paracrine effects of CDCs. We believe this addition will greatly enhance the reader's comprehension of our findings.

Reviewer's Comment 8: “Ll. 259 – 263 + ll. 358 – 383: in the context of direct cell-cell interactions vs. paracrine effects in angiogenesis, it should be noted that extracellular vesicles of cardiac progenitors indeed are able to act anti-inflammatory and proangiogenic in other studies (for example https://doi.org/10.1093/eurheartj/ehad636). This should be discussed.

Response: Thank you for your insightful comment. We have acknowledged the role of extracellular vesicles from cardiac progenitors as noted in the referenced study (https://doi.org/10.1093/eurheartj/ehad636) in our discussion.

Results, line 299: “While our focus has primarily been on cytokine-induced changes in the secretion of growth factors and cytokines, it is essential to acknowledge the multifaceted nature of CDC secretions. Numerous studies have demonstrated that CDCs extracellular vesicles promote angiogenesis and reduce inflammation in cellular experiments and animal models”

Discussion, Line 446: “In this study, we have explored the impact of cytokine pretreatment on the secretome of cardiosphere-derived cells (CDCs) and its potential implications for future clinical therapeutic strategies. While our focus has primarily been on cytokine-induced changes in the secretion of growth factors and cytokines, it is essential to acknowledge the multifaceted nature of CDC secretions and their potential relevance to cardiac regenerative medicine. Additionally, it is essential to recognize that CDCs secrete not only cytokines but also other bioactive factors, including exosomes and diuretic peptides. These secretions constitute a complex milieu that can influence the cardiac microenvironment. For instance, B-type natriuretic peptide is diuretic peptide secreted by cardiac precusor cells, has been reported to possess remarkable anti-inflammatory and anti-fibrotic properties [49-50]. It is conceivable that the collective impact of various secretions, including cytokines, exosomes and diuretic peptides, may contribute to the overall therapeutic effect of CDCs.”

We are sincerely grateful for your comprehensive review and the valuable references provided. Your suggestions have significantly contributed to the depth and breadth of our discussion on the role of CDC secretions in cardiac regeneration.

Round 2

Reviewer 1 Report

Comments and Suggestions for Authors

This is acceptable quality. Congrats!!.

Reviewer 2 Report

Comments and Suggestions for Authors

The authors addressed all of my remarks adequately. Especially the graphical abstract will greatly improve the quality of the article and the interest of the readers. From my point of view the manuscript should be accepted in the present form.